# hnRNP A1 induces aberrant *CFTR* exon 9 splicing via a newly discovered ESS element

Christelle Beaumont[1], Cristiana Stuani[2], Ming-Yuan Chou[1], Huma Shakoor[1], Maria Zlobina[1], Veronica Palaggi[1], Emanuele Buratti[2], Peter Josef Lukavsky[1]

**RNA–protein interactions play a key role in the aberrant splicing of *CFTR* exon 9. Exon 9 skipping leads to the production of a nonfunctional chloride channel associated with severe forms of cystic fibrosis. The missplicing depends on TDP-43 binding to an extended UG-rich binding site upstream of *CFTR* exon 9 3′ splicing site (3′ss) and is associated with concomitant hnRNP A1 recruitment. Although TDP-43 is the dominant inhibitor of exon 9 inclusion, the role of hnRNP A1, a protein with two RNA recognition motifs, remained unclear. In this work, we have studied the interaction between hnRNP A1 and the *CFTR* pre-mRNA using NMR spectroscopy and Isothermal Titration Calorimetry. The affinities are submicromolar, and Isothermal Titration Calorimetry data suggest complexes with a 1:1 stoichiometry. NMR titrations reveal that hnRNP A1 interacts with model *CTFR* 3′ss sequences in a fast exchange regime at the NMR timescale. Splicing assays finally show that this hnRNP A1 binding site represents a previously unknown exonic splicing silencer element. Together, our results shed light on the mechanism of aberrant *CFTR* exon 9 splicing.**

## Introduction

Cystic fibrosis is an autosomal recessive, inherited disease that affects the respiratory, digestive, and reproductive systems (Schmidt et al, 2016). The *CFTR* gene encodes for the *CFTR* protein, which acts as a chloride channel in epithelial cell membranes (Zengerling et al, 1987). The *CFTR* gene contains 27 exons. All exons included in the mRNA produce a functional protein (Kerem et al, 1989; Riordan et al, 1989; Rommens et al, 1989). The expression of most eukaryotic genes requires mRNA splicing, which removes introns and ligates exons to produce a mature mRNA. Two types of regulating, cis-acting elements are found in pre-mRNAs, which either enhance or silence splicing via interaction with specific trans-acting splicing factors. The cis-regulatory elements located upstream or downstream of the regulated exon within the pre-mRNA define which exons are included or excluded from the pre-mRNA (Wahl et al, 2009). Silencer elements interact with negative trans-acting factors such as hnRNPs (heterogeneous nuclear ribonucleoproteins) and thus repress splicing. The hnRNP family is conserved in humans, which highlights their importance for functioning in pre-mRNA maturation (Wang & Burge, 2008).

In aberrant splicing of the *CFTR* gene, there is limited inclusion of exon 9 in the final mRNA because of a polymorphic sequence in the intron upstream of this exon. The result is a defect in the nucleotide binding domain of the *CFTR* channel, which leads to *CFTR* channel dysfunction. Upstream of exon 9, in intron 8, polymorphisms can produce different variant numbers of the $(TG)_m(T)_n$ element (Dreyfuss et al, 1993; Groman et al, 2004). At the pre-mRNA level, it has been observed that polymorphic $(UG)_m(U)_n$ repeats with a low number of Us and a concomitant high number of UG repeats can repress the use of the polypyrimidine tract by altering the interaction with trans-acting proteins.

In regular splicing, the small U1 snRNA of the snRNP interacts with the 5′ss of the pre-mRNA and is supported by auxiliary factors such as the serine/arginine-rich proteins (Fig 1A). Recognition of the 3′ss is conducted by a concerted interaction of the splicing factor 1, also called branch point binding protein (SF1/BBP) to the branch point sequence and the heterodimeric auxiliary factor U2AF. Although UA2F35 recognizes the conserved AG dinucleotide at the 3′ss, U2AF65 binds the polypyrimidine tract located upstream (Matlin et al, 2005; Castellani et al, 2008; Singh & Cooper, 2012; Lee & Rio, 2015).

Studies showed that the TDP-43 protein, displaying two RRM domains and an unstructured, glycine-rich C terminus, has a high affinity for a $(TG)_{12}T_5$ sequence and binds to this site at the 3′ss of *CFTR* exon 9 (Buratti & Baralle, 2001; Buratti et al, 2001). Because this polymorphism is associated with a very short polypyrimidine tract, binding of the canonical factor U2AF65/35 is presumably impaired (Fig 1A). TDP-43, on the other hand, has a strong affinity for the concomitant, extended UG repeat and thus acts as the principal inhibitor causing aberrant splicing of *CFTR* exon 9 (Kuo et al, 2009; Lukavsky et al, 2013).

[1]CEITEC- Central European Institute of Technology, Masaryk University, Brno, Czech Republic    [2]International Centre for Genetic Engineering and Biotechnology, Trieste, Italy

Correspondence: peter.lukavsky@ceitec.muni.cz

A

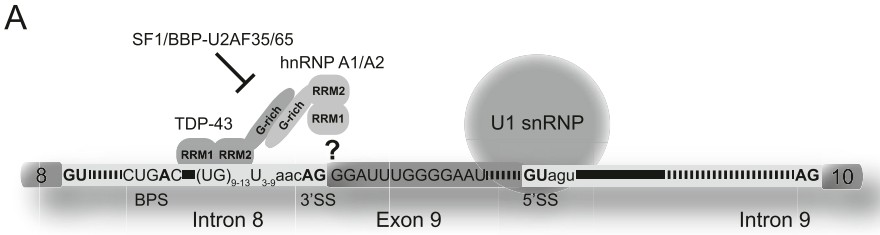

B

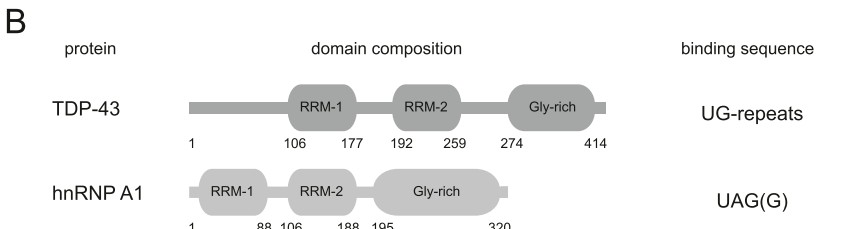

Figure 1. Aberrant splicing of *CFTR* exon 9.
**(A)** Recognition of the *CFTR* 3'ss by TDP-43 and hnRNP A1. The relevant RNA sequences in intron 8 and exon 9 are displayed. TDP-43 binds the elongated UG repeat and recruits hnRNP A1 via their C-terminal glycine-rich domains. Access of SF1/BBP and U2AF35/65 to the canonical binding sites (BPS, polypyrimidine tract, and 3'ss AG) at the 3'ss is impaired. The pre-mRNA interaction site of hnRNP A1 is unknown. **(B)** Schematic domain representation of human TDP-43 RBD and human hnRNP A1. Domain boundaries of each domain are indicated according to the full-length protein sequences (NP_031401 and NP_112420.1).

Previous studies have shown that besides the RRM domains, the C terminus of the TDP-43 protein is also essential for aberrant *CFTR* exon 9 splicing via an interaction with the C terminus of hnRNP A1 (Fig 1A and B) (Buratti et al, 2005). Using deletion mutants, Baralle, Buratti, and colleagues mapped the minimal region (the region between 321 and 366 of the TDP-43 C terminus) required for the interaction between hnRNP A1 and TDP-43 (D'Ambrogio et al, 2009). Interestingly, similar to TDP-43 the hnRNP A1 protein also contains two RRM domains and an unstructured, glycine-rich C terminus (Fig 1B). The latter provides a platform for protein–protein interactions playing a key role in regulating gene expression including alternative splicing, nuclear export from the nucleus to the cytoplasm, telomere maintenance, mRNA stability and turnover, mRNA processing, and translation regulation (Jean-Philippe et al, 2013). Although TDP-43 has been shown to be the main splicing inhibitor, hnRNP A1 is also required for aberrant *CFTR* exon 9 splicing (Buratti et al, 2005). However, until present the hnRNP A1 interaction site of its tandem RRMs at the 3'ss of exon 9 of the *CFTR* pre-mRNA remained elusive (Fig 1A).

The tandem RRMs of human hnRNP A1 are connected by a 17–amino acid linker and provide a platform for RNA interactions (Fig 1B). The tandem RRMs have been crystallized in the free and bound form where they dimerize upon binding with ssDNA. Two ssDNA sequences bind to each tandem RRM in an antiparallel manner where the 5' extremity of the ssDNA binds to the RRM1 of one tandem RRM domain, whereas the 3' extremity binds to the RRM2 of the second copy (Ding et al, 1999). The presence of this specific dimer arrangement and binding mode might be due to the packaging forces in the crystals. A study in solution using NMR spectroscopy combined with segmental isotope labeling shows that the orientation of the unbound tandem RRMs differs from the crystal structure of the free form and resembles more the one found in the ssDNA-bound form (Barraud & Allain, 2013). Another study by the Allain group presented a model of hnRNP A1 binding to the ISS-N1 pre-mRNA (Beusch et al, 2017). This model displays the opposite directionality compared to the crystal structure with ssDNA with RRM2 interacting with the 5' AG motif and RRM1 binding the 3' AG motif. However, the interaction

between the tandem RRMs and RNA still remains controversial. Despite all the structural insights from model RNA-hnRNP A1 complexes, no interaction with a naturally occurring splice site has been characterized structurally in greater detail.

Our study set out to decipher the role of hnRNP A1 in aberrant splicing of *CFTR* exon 9. Splicing assays reveal a previously unrecognized ESS within *CFTR* exon 9. NMR titration experiments and Isothermal Titration Calorimetry (ITC) measurements confirm that this element interacts with hnRNP A1 with submicromolar affinity in a specific manner forming 1:1 complexes. Our results explain how TDP-43 and hnRNP A1 work in concert to block formation of a splicing-competent complex at the *CFTR* exon 9 3'ss and thus cause aberrant *CFTR* exon 9 splicing.

## Results

### In vivo splicing assay identifies a novel ESS in *CFTR* exon

The hnRNP A1 protein binds the SELEX sequence 5'-UAGGGA/U-3' (Burd & Dreyfuss, 1994) with high affinity, and iCLIP studies of functional hnRNP A1 binding sites further indicate specificity for a 5'-UUAGGGAG-3' motif (Bruun et al, 2016). Inspection of the *CFTR* 3'ss (5'-AAC<u>AG</u>GGAUUUGGGGAAU-3') displays similar AG-rich motifs at the conserved 3'ss <u>AG</u> and within the first codons of exon 9 (Fig 1A). Binding of hnRNP A1 and A2, as well as TDP-43 in this region, is supported by the recent DeepCLIP neural network program that uses context-aware modeling to identify protein binding profiles to RNA sequences (https://deepclip-web.compbio.sdu.dk) (Gronning et al, 2020). As shown in Fig 2B, DeepCLIP correctly identifies the UG repeat sequences before 3'ss as the binding site of TDP-43 that is then closely followed by a potential, bipartite hnRNP A1 and A2 binding site. Specifically, the binding site for hnRNP A1 closely overlaps the core sequences described above (Fig 2B).

To test the effect of these motifs on splicing, we altered the RNA sequence in exon 9 while keeping its coding potential for amino

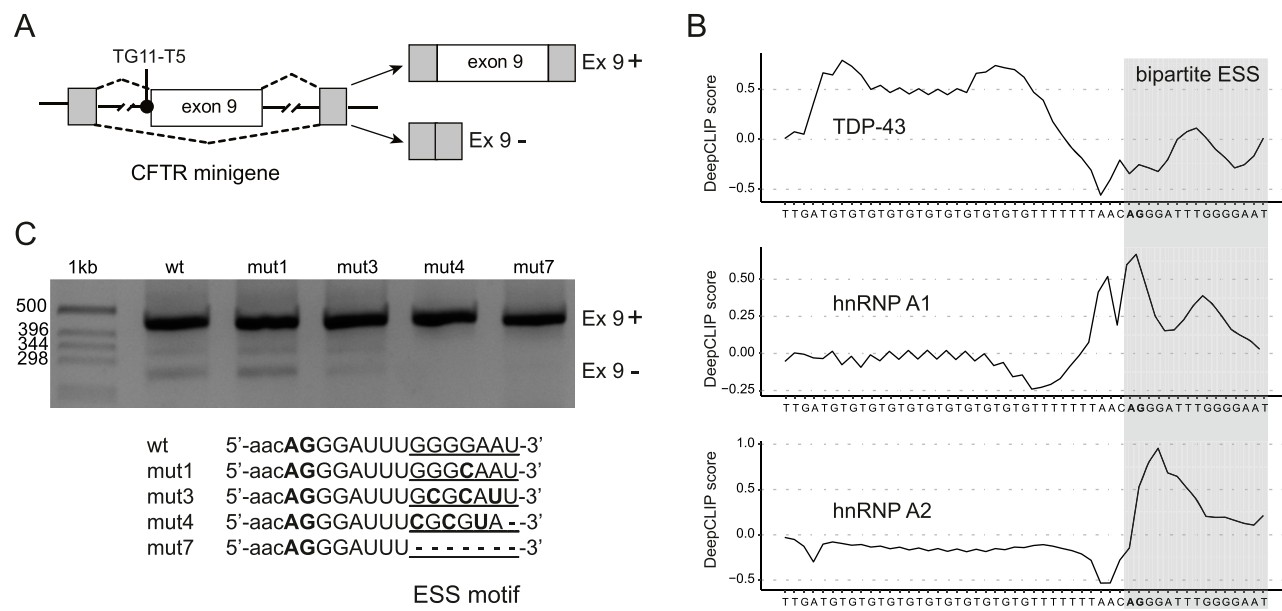

**Figure 2.   Minigene splicing assay identifies a novel ESS motif in *CFTR* exon 9.**
**(A)** Schematic representation of the *CFTR* exon 9 minigene construct used in the transfection experiments. Lines represent introns, and boxes represent exons. The intronic TG11-T5 TDP-43 binding site upstream of exon 9 is also indicated. Dashed lines indicate alternative pre-mRNA processing of the minigene, and the two major splice forms are shown on the right. **(B)** Binding sites of TDP-43, and hnRNP A1 and A2 predicted by DeepCLIP (https://deepclip-web.compbio.sdu.dk) (Gronning et al, 2020) using pretrained models from RNAcompete data sets (Ray et al, 2013). The sequence is displayed below, and the conserved AG at the 3′ss is highlighted in bold. The predicted bipartite binding site for hnRNP A1 and A1, a potential ESS motif, is shaded. **(C)** *CFTR* exon 9 inclusion upon mutation of the ESS motif in exon 9. The TG11-T5 *CFTR* exon 9 reporter minigenes (wt and mutants mut1, mut3, mut4, mut7) were transfected in HeLa cells. Progressive mutations of the ESS motif lead to full inclusion of exon 9.

acids. The universal 3′ss **AG** was not mutated because it would delete the 3′ss, but the second motif was altered progressively to delete the AG-rich motif (Fig 2C). The mutations were introduced into a *CFTR* exon 9 splicing minigene reporter system, which contains the exon 9 sequence, the splicing junctions, and part of the flanking introns with the TG11-T5 repeat at the 3′ss constituting the functional TDP-43 binding site, which is indispensable for aberrant *CFTR* exon 9 splicing as previously described (Fig 2A) (Ayala et al, 2006; Buratti et al, 2001; D'Ambrogio et al, 2009). In contrast to previous studies, the initial splicing assay was not coupled with RNAi-mediated knockdown of endogenous TDP-43 or hnRNP A1 so that the effect on splicing could be studied in a natural protein background. As an additional consideration, we opted for the TG11-T5 repeat minigene because it reflects a borderline physiological polymorphic situation in humans. Minigenes with higher TG repeats (i.e., TG13) were not considered because the resulting increased binding of TDP-43 to these repeats could have masked the inhibitory role of hnRNP A1.

The wt 5′-GGGGAAU-3′ motif and the mutant Gly-to-Gln codon change (mut1: GAA to CAA) had no effect on exon 9 exclusion (Fig 2C). In contrast, more severe disruption of the 5′-GGGGAAU-3′ motif with a change of Gly–Glu to Ala–Asp (mut3: GGGGAA to GCGCAU) almost abolished aberrant splicing of exon 9. A change of the motif to Arg–Val (mut4: GGGGAA to CGCGUA) completely abolished aberrant exon 9 splicing and had the same effect as a complete deletion (mut7) of the ESS motif (Fig 2C). Because these mutations do not affect the UG-rich TDP-43 binding site in intron 8 and because TDP-43 binding is indispensable for aberrant splicing, we conclude that the 5′-GGGGAAU-3′ motif constitutes a novel ESS in exon 9. This

motif displays a sequence similar to the consensus motif identified by SELEX (Burd & Dreyfuss, 1994) and iCLIP (Bruun et al, 2016) experiments and thus has a strong potential for hnRNP A1 binding (Fig 2B). This is also consistent with previous studies, which showed that overexpressing hnRNP A1 in a TG11-T5 minigene context was able to increase the level of exon 9 skipping (Pagani et al, 2000) and that the knockdown of hnRNP A1 leads to 100% exon inclusion because both TDP-43 and hnRNP A1 are required for exon 9 skipping (Buratti et al, 2005; D'Ambrogio et al, 2009).

To determine whether these mutants were still responsive to hnRNP A1 expression, we then overexpressed the hnRNP A1 protein according to previous studies in the fibronectin EDA exon (Muro et al, 1999). The overexpression of hnRNP A1 had the ability to almost completely inhibit exon 9 inclusion in the *CFTR* wt minigene (Fig 3A). However, this ability was progressively impaired when hnRNP A1 was overexpressed with the mut3 and mut4 *CFTR* minigenes, demonstrating that these mutations had successfully impaired the A1 binding site (Fig 3A and B).

Finally, to further determine whether hnRNP A1 could act as splicing silencer proteins of *CFTR* exon 9, we silenced both hnRNP A1 and A2 proteins in the presence of a *CFTR* exon 9 minigene, which also carried the C155T artificial variant. This variant was designed in previous studies to display a higher level of exon 9 skipping (~50%) in order to better appreciate eventual changes in exon 9 inclusion in both directions (Pagani et al, 2003) and is therefore ideal to test for variations in splicing factor expression levels. The reason why hnRNP A1 and A2 had to be silenced together is due to the fact that both proteins are well known to compensate for each other in many pre-mRNA splicing events including *CFTR* exon 9 (D'Ambrogio et al,

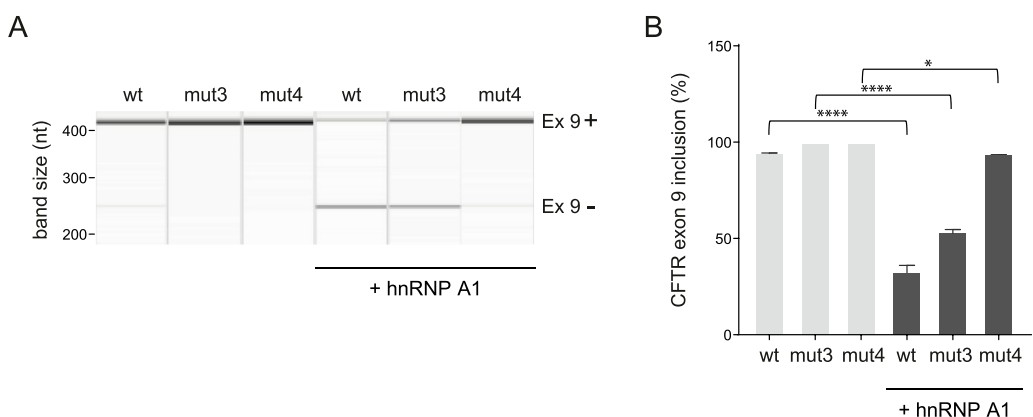

**Figure 3. Overexpression of hnRNP A1 protein in the presence of the *CFTR* exon 9, mut3, and mut4 minigenes.**
**(A)** *CFTR* exon 9 reporter minigenes (wt and mutants mut3 and mut4) were transfected in HeLa cells with and without hnRNP A1 overexpression. Band sizes are indicated and correspond to inclusion of *CFTR* exon 9 (upper band) and exclusion (lower band). **(B)** Summarized results from three biological replicates (four individual transfection experiments); one-way ANOVA was performed, and data are shown as the mean ± SD.

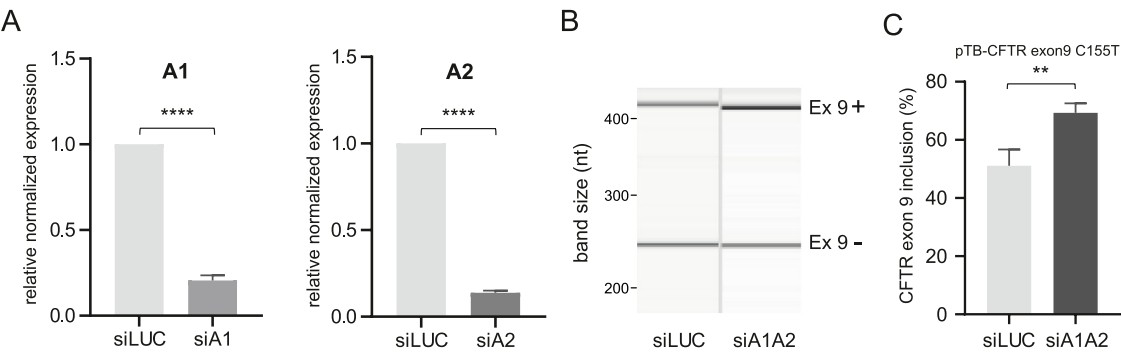

**Figure 4. Effects of hnRNP A1 and A2 knockdown on *CFTR* exon 9 inclusion, using the pTB-*CFTR* exon 9 C155T plasmid (Pagani et al, 2000).**
**(A)** Two graphs report the level of hnRNP A1 and hnRNP A2 mRNA knockdown with siRNAs, siA1 and siA2, respectively, or control siRNA (siLUC) as determined by RT–qPCR analysis. Three independent experiments were plotted in a column graph, and an unpaired *t* test was performed using GraphPad software. **(B)** Representative gel obtained using QIAxcel High Resolution Kit (QIAGEN) showing the inclusion of CFTR exon 9 (upper band) and exclusion (lower band) after treatment with hnRNP A1/A2 siRNA (siA1A2) and control siRNA (siLUC). **(C)** Quantification of CFTR exon 9 inclusion from three independent experiments. The quantification was performed using QIAxcel High Resolution Kit software (QIAGEN), and the single values were plotted in a column graph: an unpaired *t* test was performed using GraphPad software.

2009). Accordingly, when we just silenced hnRNP A1 in the presence of hnRNP A2 no effect could be observed (data not shown). As shown in Fig 4, however, the efficient silencing of hnRNP A1 and hnRNP A2 in HeLa cells resulted in a significant increase in *CFTR* exon 9 inclusion (Fig 4A–C). Thus, our silencing experiments further support the presence of a novel, bipartite ESS in exon 9, which is sensitive to the presence of proteins hnRNP A1 and A2.

**Binding studies of hnRNP A1 tandem RRMs with the novel *CFTR* exon 9 ESS sequence**

Next, we aimed to characterize the interaction between hnRNP A1 and the *CFTR* exon 9 ESS sequence identified by our splicing assay. We used a ssDNA sequence comprising the *CFTR* exon 9 ESS sequence and characterized the interaction using ITC measurements. The hnRNP A1 tandem RRM protein was titrated with 5'-CAGGGATTTGGGGAC-3' ssDNA sequence (Fig 5A). The ssDNA at a concentration of 200 $\mu$M was added stepwise to a 28 $\mu$M protein solution. The stoichiometry represented by the N parameter is 0.736, indicating that one protein molecule binds to

one molecule of ssDNA. The value of the dissociation constant ($K_d$ = 233 nM) reveals a strong affinity between the protein and the ESS motif. We note that the reaction is spontaneous with a favorable enthalpy and entropy. The enthalpy value $\Delta H$ = −2.37 × $10^4$ kcal/mol is consistent with an exothermic reaction. The negative entropy value ($\Delta S$ = −56.6 kcal/mol) indicates that the reaction weighs toward the formation of a more ordered structure such as an RNA–protein complex (Fig 5A). Next, we set out to investigate the role of the individual RRMs of hnRNP A1. The RRM1 domain of the hnRNP A1 protein is titrated with the 5'-CAGG-GATTTGGGGAC-3' ssDNA sequence (Fig 5B). The protein concentration in the cell is 25 $\mu$M, and the ssDNA concentration is 200 $\mu$M. The N value of 0.496 is indicative of two protein molecules binding to one ssDNA, which is in line with two binding sites in the ESS motif. The $K_d$ of 5.8 $\mu$M reveals a weaker affinity as compared to the tandem RRMs. The RRM2 domain of the hnRNP A1 protein was also titrated with the 5'-CAGG-GATTTGGGGAC-3' ssDNA sequence (Fig 5C). The protein (25 $\mu$M) is in the cell and the ssDNA (200 $\mu$M) is again titrated to the protein solution. The N value of 0.452 also points toward two protein molecules binding to the two ESS binding sites. The $K_d$ = 6.8 $\mu$M also indicates a weak affinity. The fact that both isolated RRMs bind the ESS sequence with

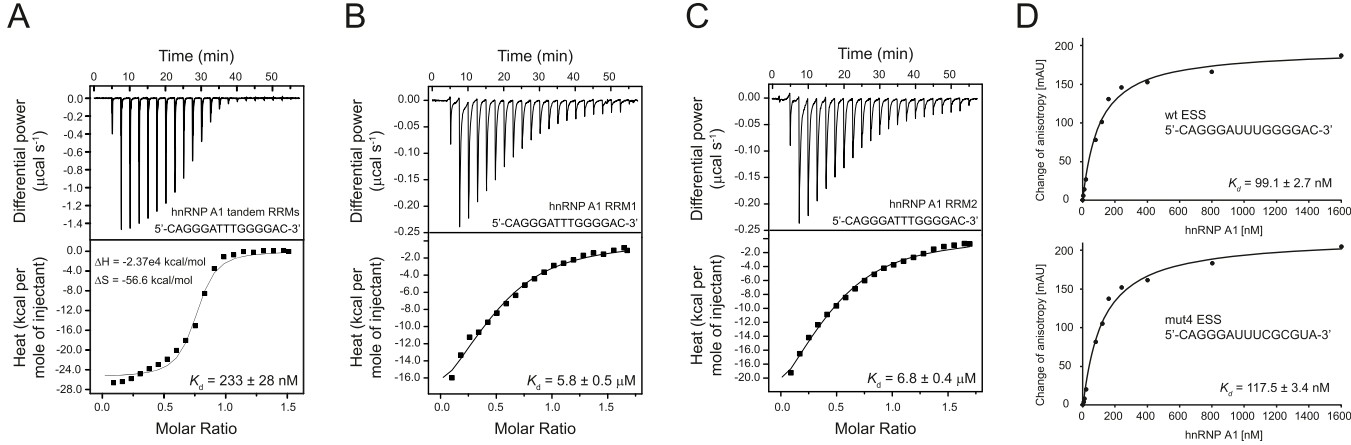

**Figure 5. hnRNP A1 binds the novel bipartite ESS motif in *CFTR* exon 9.**
**(A)** Affinity of the hnRNP A1 tandem RRMs for the bipartite ESS motif was determined by Isothermal Titration Calorimetry (ITC). The determined $K_d$ value is indicated with SD, as well as $\Delta H$ and $\Delta S$. **(B)** Affinity of hnRNP A1 RRM1 for the bipartite ESS motif was determined by ITC. Two copies of hnRNP A1 RRM1 bind to the bipartite ESS motif. The determined $K_d$ value is indicated with SD. **(C)** Affinity of hnRNP A1 RRM2 for the bipartite ESS motif was determined by ITC. Two copies of hnRNP A1 RRM2 bind to the bipartite ESS motif. The determined $K_d$ value is indicated with SD. **(D)** Affinity of the hnRNP A1 tandem RRMs for the bipartite wt (top) and mut4 (bottom) ESS motif was measured by fluorescence anisotropy. The determined $K_d$ value represents the average of three individual measurements and is indicated with SD.

micromolar affinity as compared to the tandem RRMs, which bind in the submicromolar range, is consistent with a cooperative binding mode of the tandem RRMs previously described for other ESS sites (Fig 5A–C) (Okunola & Krainer, 2009).

We also attempted to measure ITC data using both single RRMs and tandem RRMs of hnRNP A1 and the corresponding ssRNA 5′-CAGGGAUUUGGGGAC-3′ comprising the bipartite ESS motif. The data confirmed the interaction with ssRNA, but the systems were more complex than with the corresponding ssDNA molecules. We titrated the ssRNA with proteins and vice versa with the similar outcome (data not shown). We speculate that this might be due to the presence of a strong tertiary structure in the ssRNA (e.g., quadruplex formation), which needs to be melted upon protein binding and thus hampers the fitting of the ITC data to simple one-site or two-site binding models. We therefore performed fluorescence anisotropy measurements to confirm the binding to the bipartite ESS motif using fluorescein-labeled ssRNA. The wt ESS ssRNA bound the tandem RRMs of hnRNP A1 with similarly strong affinity ($K_d$ = 99 nM) as the corresponding ssDNA (Fig 5D). The mut4 ESS motif ssRNA, on the other hand, displayed a consistently weaker interaction ($K_d$ = 117 nM) with the tandem RRMs of hnRNP A1 as compared to the wt ssRNA, which is in agreement with the impairment of the hnRNP A1 binding site (Figs 5D and 3). The fact that only a small difference in affinity between wt and mut4 ESS motif ssRNA but a strong difference in splicing efficiency was observed might reflect that in vitro experiments can only partially mimic the complex interaction network and possibly thereby induced directionality of hnRNP A1 binding at the 3′ss of *CFTR* exon 9.

### Binding studies of the individual hnRNP A1 RRMs with the separated ESS motifs

To further investigate the interaction of hnRNP A1 with the ESS sequence, we divided the ssDNA into two segments. We measured the affinities for both parts of the ESS motif (5′-CAGGGAT-

3′ and 5′-TGGGGAAT-3′). The ITC experiments show that the binding of RRM1 for 5′-CAGGGAT-3′ and 5′-TGGGGAAT-3′ ssDNA occurs at a 1:1 ratio with $K_d$ values of 0.85 and 7.25 $\mu$M, respectively (Fig 6A and B).

Thus, the RRM1 domain has an almost 10-fold higher affinity for the first part of the bipartite ESS motif. For RRM2, the ITC data show that the binding of 5′-CAGGGAT-3′ and 5′-TGGGGAAT-3′ DNA occurs also at a 1:1 ratio with a $K_d$ value of 0.58 and 3.05 $\mu$M, respectively (Fig 6C and D). In conclusion, the RRM2 domain has a fivefold higher affinity for the first part of the ESS motif as well. Nonetheless, the similar binding affinities of the individual RRM domains for the separated ESS motifs do not allow to reach a conclusion about the orientation of the ESS motif upon interaction with the tandem RRM scaffold of the hnRNP A1 protein.

### Assessment of the ESS-hnRNP A1 interaction by solution NMR spectroscopy

We used the assignment of the hnRNP A1 protein published by the Allain group and superimposed them onto our $^1$H-$^{15}$N HSQC spectra of the amide fingerprint region of hnRNP A1 (Barraud & Allain, 2013). We produced four complexes comprising each separated RRM domain with each of the separated ESS segments.

The $^1$H-$^{15}$N HSQC spectra of the four complexes were recorded at 298K. Because the second ESS segment contains four guanines, which could form a G-quadruplex irrelevant to exon 9 splicing, and because previous studies showed that G-quadruplexes bind to the hnRNP A1 protein with high affinity (Liu et al, 2017), we mutated the second ESS motif from 5′-TGGGGAAT-3′ to 5′-CTGGGCAC-3′ according to the mut1, which had no effect on aberrant splicing (Fig 2B).

With the titration results, we could confirm the binding of the RRM1 domain with the 8-mer ssDNA model sequence (5′-CAGG-GATC-3′) (Fig 6A and B). The ssDNA binding of the free protein shown in blue can be followed upon stepwise addition of the ssDNA

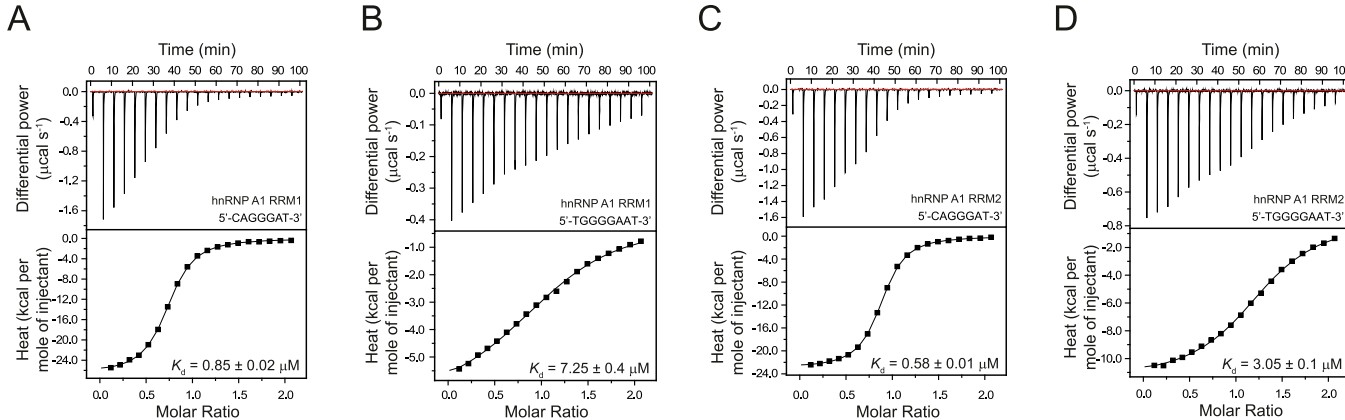

**Figure 6. Both hnRNP A1 RRM1 and RRM2 bind the individual segments of the ESS motif of *CFTR* exon 9.**
**(A, B)** Affinity of hnRNP A1 RRM1 for the individual segments of the ESS motif was determined by Isothermal Titration Calorimetry. 1:1 complexes are formed with the individual segments of the ESS motif. The determined $K_d$ values are indicated with SD. **(C, D)** Affinity of hnRNP A1 RRM2 for the individual segments of the ESS motif was determined by Isothermal Titration Calorimetry. 1:1 complexes are formed with the individual segments of the ESS motif. The determined $K_d$ values are indicated with SD.

(Fig 7A). The chemical shift perturbations (CSPs) observed at each titration point indicate the potential ssDNA binding site on the protein and that the interaction occurs in a fast exchange regime in the NMR timescale. We considered all the CSPs > 0.06 more than average, indicating that the affected amino acids are either involved in the interaction with the 8-mer ssDNA model or indirectly affected by ssDNA binding. The affected amino acids are Gln13, Leu17, Gly20, Phe24, Glu29, Ser30, Thr42, Val45, Val46, Thr52, Ser55, Gly57, Ala72, Asn74, His78, Lys88, Ala90, Val91, Ser92, and Arg98 (Fig 7B). Mapping these CSPs onto the structure shows that the interaction site of the ssDNA mostly comprises the beta-sheet surface of the protein, which is the canonical RNA interaction site on the RRM fold.

We subsequently recorded $^1$H-$^{15}$N HSQC spectra (298K) to monitor the binding of the RRM1 domain with the 8-mer DNA model (5'-CTGGGCAC-3') corresponding to the second part of the ESS motif (Fig 7C). We used the same ssDNA titration steps as for the first ESS motif to monitor ssDNA binding. Again, the nature of the CSPs indicated a fast exchange regime at the NMR timescale that the RRM1 domain binds also to the second part of the ESS motif. All the CSP > 0.06 are considered more than average and involve amino acids Gln13, Arg15, Lys16, Leu17, Phe18, Val45, Met47, Arg56, Gly57, Gly59, Phe60, Ala64, Asn74, His78, Lys79, Arg83, Val84, Val85, Arg89, Ala90 (Fig 7D). The interaction site of the ssDNA comprises mostly the beta sheet of the protein, which is again the canonical RNA interaction sites on the RRM fold.

Next, we monitored binding of RRM2 to the separate ESS motifs. Interaction occurs in a fast exchange regime in the NMR timescale. The NMR titration confirmed the interaction between RRM2 and the first ESS motif (5'-CAGGGATC-3'), as well as the second ESS motif (5'-CTGGGCAC-3') (Fig 7A and C). Thus, our NMR titration experiments confirm the ITC results, which showed that both individual RRMs interact with each part of the ESS motif. The CSPs > 0.05 observed upon binding of RRM2 to the 8-mer ssDNA (5'-CAGGGATC-3') involve amino acids Leu103, Thr104, Lys106, Lys107, Ile108, Val110, Lys131, Glu136, Thr139, Asp140, Gly142, Gly144, Phe151, Ser159, Asp161, His169, Lys180, Gln185, Met187, Ser189 (Fig 8A and B).

Likewise, the CSPs > 0.05 observed upon binding of RRM2 and the 8-mer ssDNA (5'-CTGGGCAC-3') involve a similar region on the protein (Arg98, Thr104, Val105, Lys106, Lys107, Val110, Gly112, Glu136, Ile137, Met138, Thr139, Asp140, Gly142, Ser143, Arg147, Phe151, Val152, Val178, Arg179, Lys180, Ala181, Leu182, Gln185, Met187, Ala188, Ser189, Ala190) (Fig 8C and D). The interaction sites of both ssDNAs comprise mostly the beta sheet of the protein and the C-terminal helix.

Next, we used the CSP results from the NMR titration experiments of the individual RRMs and the separated ESS segments to visualize the potential ESS binding site on the hnRNP A1 tandem RRM structure determined by the Allain group (Barraud & Allain, 2013). The overall fold and arrangement of the tandem hnRNP A1 RRMs present a continuous binding site for the novel, bipartite *CFTR* exon 9 ESS segment (Fig 9). Despite the fact that our experiments did not reveal a preference of one of the RRMs for a certain segment of the bipartite ESS sequence, both possible orientations of the ESS segment on the tandem RRMs cover the same continuous protein surface and could thereby impair access of canonical splicing factors to the *CFTR* exon 3'ss and subsequently cause aberrant *CFTR* exon 9 splicing.

## Discussion

Alternative splicing is an important cornerstone of RNA-based, posttranscriptional regulation of gene expression. Production of diverse protein products through inclusion and exclusion of certain exons from the same pre-mRNA is a major contributor to the complexity of higher organisms. But at the same time, splicing also depends on correct RNA and protein products for proper functioning, and thus, variations in both components can lead to missplicing and disease (Cooper et al, 2009). One such variant is found in the polymorphic $(TG)_m(T)_n$ locus upstream of *CFTR* exon 9 within intron 8, which leads to recruitment of TDP-43 to the pre-mRNA and subsequent TDP-43–dependent recruitment of hnRNP A1. The UG-rich binding site of TDP-43 in the *CFTR* pre-mRNA is well

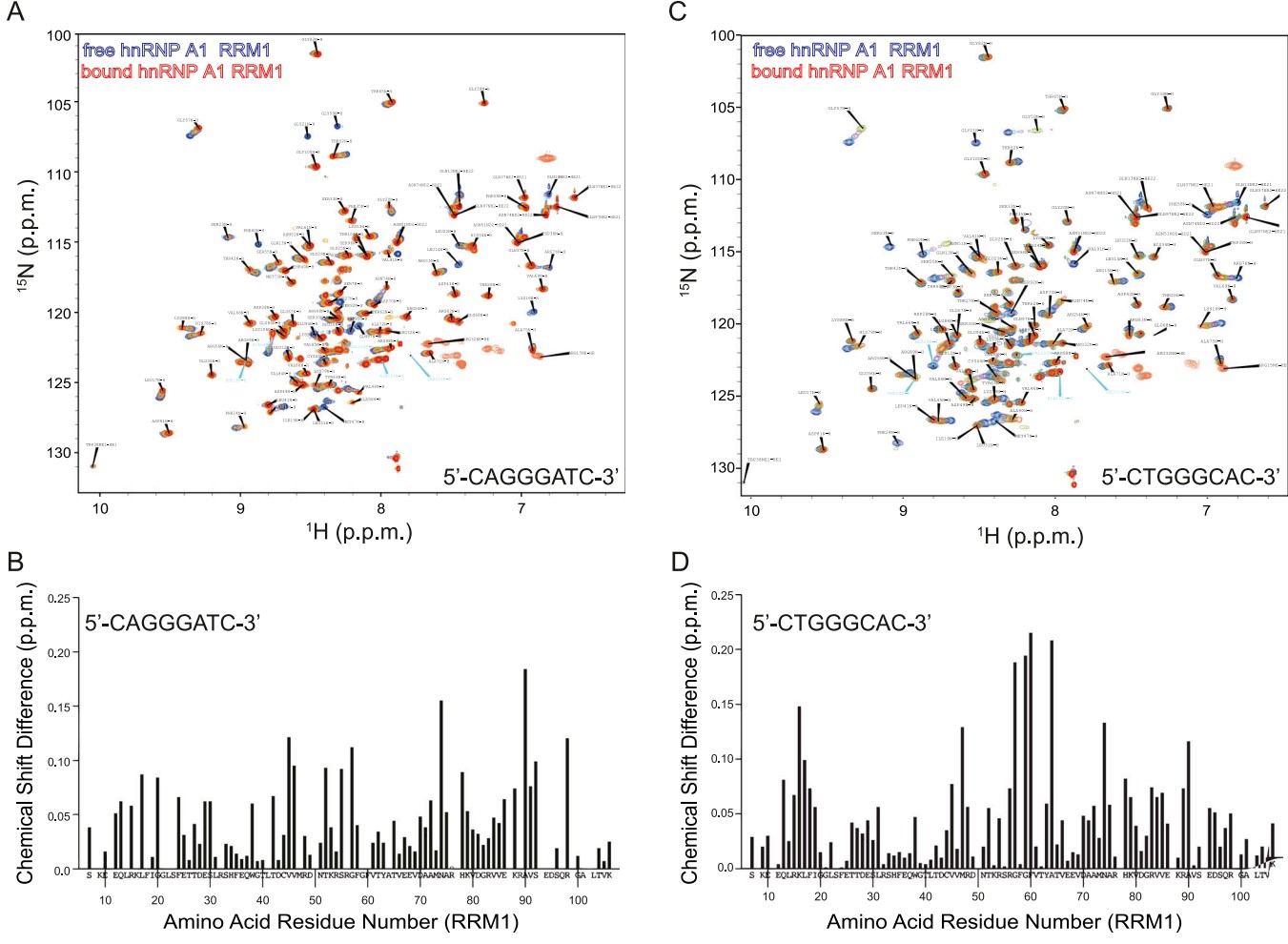

**Figure 7. NMR titration of hnRNP A1 RRM1 reveals specific interactions with the individual segments of the ESS motif.**
**(A, C)** Superimposition of $^1$H-$^{15}$N HSQC spectra of $^{15}$N-labeled hnRNP A1 RRM1 in the free (blue) and short ESS motif-bound form (red). The amide cross-peaks shifted upon interaction are labeled with amino acid type and residue number. **(B, D)** Combined chemical shift perturbations $\Delta\delta = [(\delta H^N)^2 + (\delta N/6.51)^2]^{1/2}$ of hnRNP A1 RRM1 amide resonances upon short ESS motif binding versus the hnRNP A1 RRM1 amino acid sequence.

characterized both functionally (Buratti et al, 2001; Ayala et al, 2006) and structurally (Kuo et al, 2009; Lukavsky et al, 2013), and it is known that hnRNP A1 is recruited to the 3'ss via a protein–protein interaction with TDP-43 (Buratti et al, 2005; D'Ambrogio et al, 2009). The molecular origin of *CFTR* exon 9 splicing inhibition via TDP-43–dependent recruitment of hnRNP A1, on the other hand, remained elusive.

Our study identifies a new ESS comprising a bipartite sequence motif at the *CFTR* exon 9 3'ss and the 5' end of exon 9 (5'-AGG-GAUUUGGGGAAU-3'). We show that hnRNP A1 is able to specifically interact with this bipartite ESS motif and that the integrity of this site is instrumental for aberrant splicing of *CFTR* exon 9. Our minigene splicing assay shows that the wt sequence around the *CFTR* 3'ss produces several bands corresponding to the inclusion and skipping of exon 9. Four different mutations in exon 9, which alter the 5'-GGGGA-3' motif but leave the 3'ss intact, clearly show that variation of this sequence affects the inclusion of exon 9 in the mRNA. Mutating the 5'-GGGGA-3' motif to 5'-CGCGT-3' and its complete deletion are associated with nonskipping of exon 9, that

is, the full inclusion into the final mRNA and loss of the ability to respond to hnRNP A1 overexpression. This is further supported by minigene splicing assays under hnRNP A1 and A2 knockdown, which show two significant exon 9 inclusions upon depletion of the two proteins (Fig 4C). Likewise, hnRNP A1 overexpression leads to complete *CFTR* exon 9 skipping only the wt ESS motif, whereas mutant ESS motifs impair *CFTR* exon 9 skipping. Thus, these experiments reveal for the first time a bipartite ESS motif in *CFTR* exon 9 causative for exon 9 skipping.

We also investigated the RNA binding mode of hnRNP A1 to the ESS that silences splicing of *CFTR* exon 9. The ITC, NMR titration experiments, and fluorescence anisotropy measurements strongly suggest that both RRMs of hnRNP A1 interact with the bipartite ESS motif and that affinity drops when this ESS motif is mutated (Fig 5). The affinities obtained from ITC experiments using the isolated RRMs and single binding sites of the bipartite ESS motif are comparable, suggesting that there is no preferred sequence motif for the individual RRMs determining the orientation of hnRNP A1 on the pre-mRNA target. Previous studies have shown that the hnRNP

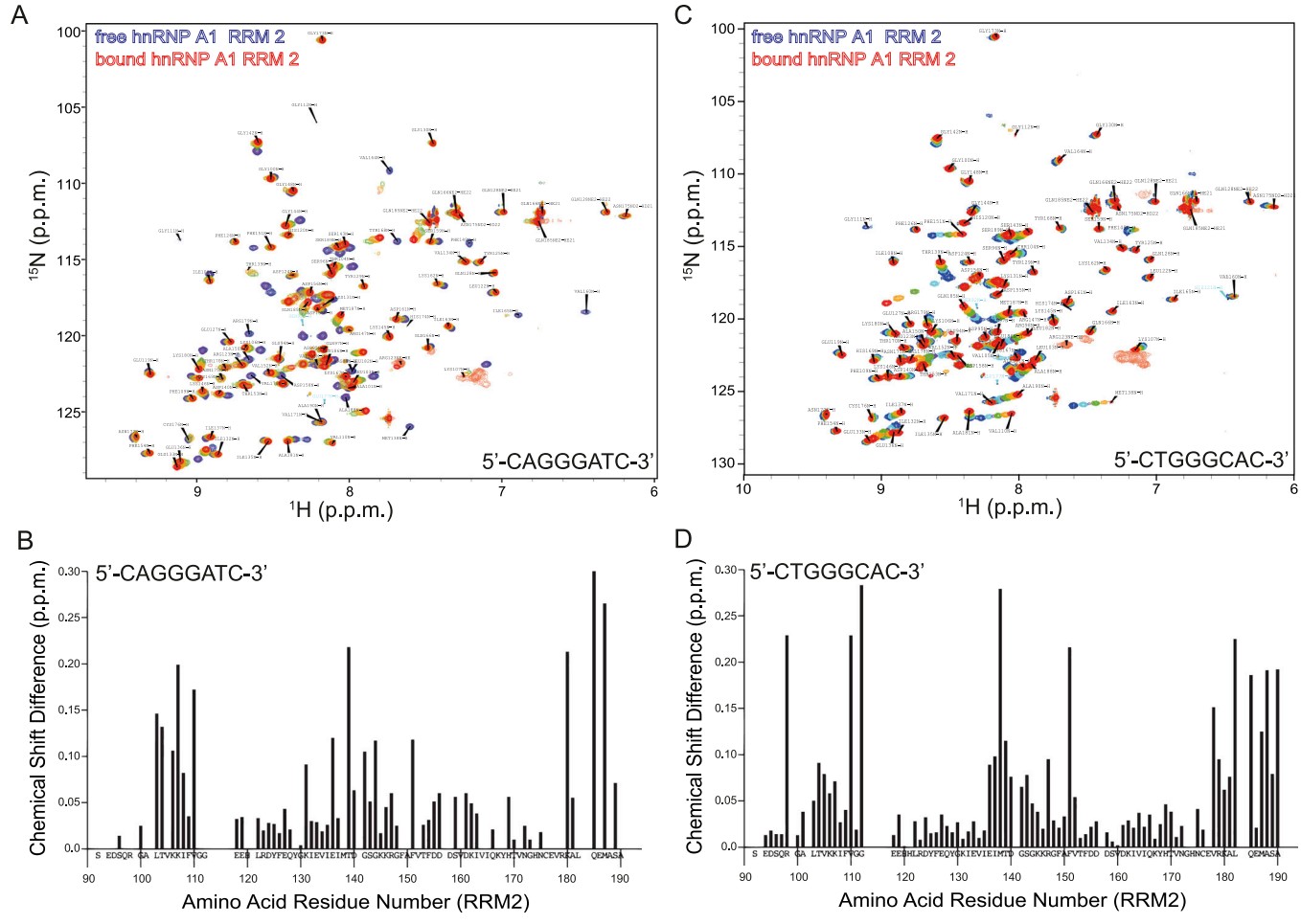

**Figure 8. NMR titration of hnRNP A1 RRM2 reveals specific interactions with the individual segments of the ESS motif.**
**(A, C)** Superimposition of $^1$H-$^{15}$N HSQC spectra of $^{15}$N-labeled hnRNP A1 RRM2 in the free (blue) and short ESS motif-bound form (red). The amide cross-peaks shifted upon interaction are labeled with amino acid type and residue number. **(B, D)** Combined chemical shift perturbations $\Delta\delta = [(\delta H^N)^2 + (\delta N/6.51)^2]^{1/2}$ of hnRNP A1 RRM2 amide resonances upon short ESS motif binding versus the hnRNP A1 RRM2 amino acid sequence.

A1 protein can bind specifically and with high affinity to ssRNA or ssDNA sequences because there are no qualitative differences between them (Ding et al, 1999) and several high-affinity sequences have also been discovered for hnRNP A1. Indeed, SELEX experiments isolated a high-affinity RNA sequence of 5′-UAGGGA/U-3′ that is recognized by hnRNP A1 (Burd & Dreyfuss, 1994), which is similar to the *CFTR* exon 9 ESS motif. The specific recognition of the motif was confirmed by an x-ray crystallography structure of hnRNP A1 tandem RRMs bound to a telomeric ssDNA sequence (5′-TTAGGGTTAGGG-3′) (Ding et al, 1999). The structure revealed that both RRM1 and RRM2 of the same monomer interact with two different strands of ssDNA, in an antiparallel manner. In this structure, RRM1 binds to four nucleotides (5′-TAGG-3′), whereas RRM2 preferably binds to five nucleotides (5′-TTAGG-3′). The *CFTR* exon 9 ESS motif also contains two binding sites with similar sequence, but our NMR and ITC data suggest that a single copy of hnRNP A1 binds both motifs via the tandem RRMs. Thus, the dimerization in the x-ray structure might be just caused by crystal packing forces and not be relevant for *CFTR* exon 9 skipping.

A study performed by Allain and colleagues has identified optimal recognition motifs that bind the individual RRMs (Beusch et al, 2017). RRM1 binds to the sequence 5′-U/CAGG-3′ and RRM2 to the sequence 5′-U/CAGN-3′. They also show that both RRMs of hnRNP A1 bind to their RNA target containing the two AG binding sites without dimerization of the protein. They also presented a structural model of hnRNP A1 binding the ISS-N1 pre-mRNA with opposite directionality where RRM2 interacts with the 5′ motif and RRM1 binds the 3′ motif.

Another study investigated the interaction between hnRNP A1 and an RNA target, namely, the pri-mir-18a. Again, this interaction involves both RRMs and a region encompassing two 5′-UAG-3′ motifs (Kooshapur et al, 2018). In this study, it was demonstrated that cooperative binding of both domains to the related RNA motifs results in a strong enhancement of binding affinity and allows the unwinding of the RNA target as a stem–loop. Our ITC experiments also reveal that individual RRMs interact with the ESS in the low micromolar range, whereas the tandem RRMs display nanomolar affinity consistent with cooperative binding of both RRMs to the bipartite ESS.

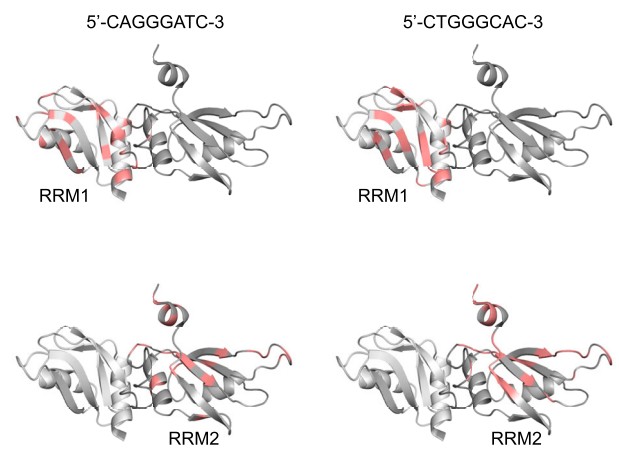

**Figure 9. hnRNP A1 tandem RRMs display a continuous binding site for the novel *CFTR* exon 9 ESS motif.**
The CSPs from the titration experiments of the individual hnRNP A1 RRMs and the short segments of the ESS motif are displayed on the structure of the hnRNP A1 tandem RRMs (PDB ID: 2lyv). RRM1 is colored in light gray and RRM2 in dark gray. The CSPs are indicated on the ribbon structure in salmon, and the sequence of the corresponding ESS segments is displayed on top.

Further work is needed to determine the orientation of hnRNP A1 along the *CFTR* exon 9 ESS element. Binding this natural target with directionality might require the protein–protein interaction of full-length TDP-43 and hnRNP A1 mediated by their glycine-rich C-terminal domains (Buratti et al, 2005; D'Ambrogio et al, 2009). Consistent with this idea, disrupting this interaction via mutations or deletion in the C termini of TDP-43, as well as hnRNP A1, abolishes exon 9 skipping despite the fact that hnRNP A1 maintains its ability to bind to the target *CFTR* exon 9 ESS element. Studying the RNA–protein and protein–protein interactions within the entire aberrant splice site using full-length TDP-43 and hnRNP A1 is required to shed light on the precise mechanism of aberrant splicing occurring at *CFTR* exon 9.

# Materials and Methods

### Cloning, expression, and purification of hnRNP A1 tandem RRMs

The sequence of DNA encoding the tandem RRMs composed of 196 amino acids was subcloned by PCR amplification into the pET28a vector. The construct contains a 2xHis$_6$ tag, a lipoyl domain followed by a TEV protease cleavage site for the tag, and the lipoyl domain to be removed after the first step of purification. Recombinant proteins were overexpressed in *E. coli* BL21 (DE3) codon plus cells (Novagen) in LB-rich or M9 minimal media supplemented with $^{15}NH_4Cl$ or $^{15}NH_4Cl$ and $^{13}C$-glucose. The cells were grown at 37°C to OD600 ~0.6 and then at 20°C for 1 h. Protein expression was induced at OD600 ~0.9 by the addition of 1 mM isopropyl-β-D-thiogalactopyranoside (IPTG) and further incubated for 15 h at 20°C. Cells were harvested after 15 h and then centrifuged at 4,000*g* for 20 min at 4°C. The cell pellet was resuspended in lysis buffer A

(20 mM Hepes, pH 7.5, 1 M NaCl, 30 mM imidazole, 5 mM BME, 10% [wt/vol] glycerol) supplemented with protease inhibitors, and lysed by high pressure with a French press. The cell lysate was centrifuged for 30 min at 39,000*g* at 4°C. The supernatant was loaded on a Ni-NTA column on an ÄKTA Prime purification system (GE Healthcare), and after washing with buffer A, the protein of interest was eluted with 300 mM imidazole gradient. The fractions containing the protein of interest were collected, and then, TEV protease was added according to the estimation of the amount of the protein and then dialyzed against buffer A without imidazole overnight at RT to perform the cleavage of tag. Then, the sample was reloaded onto a Ni-NTA column to remove the tag, TEV protease, and uncleaved protein. Samples containing the protein were further purified by size-exclusion chromatography on a Superdex 75 column equilibrated with NMR buffer (50 mM sodium phosphate, pH 6.5, 1 mM BME, 50 μM EDTA) at 4°C and concentrated to 1 mM with Vivaspin 10.000 MWCO.

### Cloning, expression, and purification of RRM1 and RRM2 of hnRNP A1

The sequence of DNA encoding RRM1 and RRM2 individually (RRM1 from residues 1–105 and RRM2 from residues 91–196) was subcloned by PCR amplification into the same modified pET28a vector as above. Recombinant proteins were overexpressed and purified as for the construct containing both RRMs. Samples containing proteins were further purified by size-exclusion chromatography on a Superdex 75 column equilibrated with NMR buffer (25 mM sodium phosphate, pH 6.5, 1 mM BME, 50 μM EDTA) at 4°C and concentrated to 1 mM with Vivaspin 10.000 MWCO.

### ssDNA purification and desalting and ssRNA sample preparation

The ssDNAs and 3′ end fluorescently labeled ssRNAs were purchased from Dharmacon. The 15-mer ssDNA (5′-CAGGGATTTGGG-GAC-3′), the 7-mer ssDNAs (5′-CAGGGAT-3′), and the three 8-mer ssDNAs (5′-TGGGGAAT-3′, 5′-CAGGGATC-3′, and 5′-CTGGGCAC-3′) were purified under denaturing conditions using the UltiMate 3000 HPLC system with anion-exchange preparative DNAPac PA100 Nucleic Acid Column (both from Thermo Fisher Scientific) as described before (Zlobina et al, 2016). We performed the purification at 85°C with a flow rate of 20 ml/min. The column was equilibrated with the buffer containing 6 M urea, 12.5 mM Tris–HCl, pH 7.4. The sample was loaded into the sample loop and then eluted from the column with the buffer containing 6 M urea, 12.5 mM Tris–HCL, 500 mM NaClO$_4$, pH 7.4, up to 50% of NaClO$_4$ while collecting 10 ml fractions. The fractions were analyzed by SDS–PAGE (20% acrylamide, 8 M urea). The ssDNA was visualized with 0.1% toluidine. The fractions containing the ssDNA were desalted using a ÄKTA Prime FPLC system (GE Healthcare) equipped with three 5-ml DEAE weak anion-exchange columns in series (Easton et al, 2010). The columns were equilibrated with the buffer containing 20 mM ammonium bicarbonate, pH 7.5 (degassed for 30 min). The sample was diluted twice, then loaded into the sample loop, and injected with the buffer at a flow rate of 5 ml/min. A steep gradient containing 100% of 2.53 M ammonium bicarbonate, pH 8, was applied to collect 10 ml fractions. Fractions with the ssDNA were collected and then

lyophilized. The 15-mer wt (5′-CAGGGAUUUGGGGAC-3′′-FL) and mutant (5′-CAGGGAUUUCGCGUA-3′′-FL) ssRNAs were deprotected according to the manufacturer's instructions, then lyophilized, and dissolved in NMR buffer for FA measurements.

### NMR experiments

All NMR experiments were carried out using Bruker Avance III HD 700-, 850-, and 950-MHz spectrometers each equipped with cryoprobes. Data acquisition was performed at 298K, and samples were measured in NMR buffer with 10% $D_2O$. Data were processed using Topspin 3.2/3.5 (Bruker) and analyzed with Sparky (http://www.cgl.ucsf.edu/home/sparky/). Protein resonance assignments were taken from BioMagResBank under the accession number 18728 (Barraud & Allain, 2013). The buffer used (25 mM sodium phosphate, pH 6.5, 1 mM DTT) was similar to our NMR buffer (see above) so that resonance assignments could be directly transferred to our spectra. CSPs from NMR titration experiments (2D $^1$H-$^{15}$N HSQC spectra) were calculated using the formula: $\Delta\delta = [(\delta H^N)^2 + (\delta N/6.51)^2]^{1/2}$. HSQC titrations were carried out using $^{15}$N-labeled hnRNP A1 protein constructs and unlabeled ssDNA. Spectra were recorded at the following molar ratios (protein/ssDNA): 1/0.2; 1/0.4; 1/0.6; 1/0.8; 1/1; 1/2.

### ITC measurements

ITC experiments were performed on a VP-ITC instrument (MicroCal) at 25°C. The calorimeter was calibrated according to the manufacturer's instructions. Protein and ssDNA samples were dialyzed against the NMR buffer. Concentrations of proteins and ssDNAs were determined using optical absorbance at 280 and 260 nm, respectively. The ssDNA (200 $\mu$M) was injected into the sample cell containing the protein at a concentration of around 20 $\mu$M. Titrations consisted of 20 injections of 2 $\mu$l except for the first injection (1.5 $\mu$l) with a 2-min spacing between each injection. Data were fitted with the one-site binding model using MicroCal Origin.

### Fluorescence anisotropy measurements

Fluorescence anisotropy was measured on Tecan Microplate Reader Infinite F500 (Tecan) equipped with a plate reader using 96-well plates. All measurements were performed in NMR buffer at 35°C. Measurements were performed in 10 $\mu$l reaction volume in which 10 nM 3′ end fluorescein-labeled ssRNA (wt or mutant) was titrated with hnRNP A1 tandem RRMs. The data points represent the average of three measurements. The fitting was performed with Origin software (OriginLab) (Pollard, 2010). All data were normalized for adequate visualization.

### Splicing assay

Plasmid TG11-T5 has been previously described by Niksic et al (1999). Mutations were inserted using complementary nucleotides, and liposome-mediated transfections of 3 x 10$^5$ human HeLa cells were performed using Effectene (QIAGEN) according to the manufacturer's instructions. After 18 h from the transfection, the RNA was purified using the RNeasy kit (QIAGEN) according to the

manufacturer's instructions, and the RT–PCRs to specifically amplify the minigene transcripts were performed as previously reported (Pagani et al, 2000). For the hnRNP A1 and A2 silencing experiments, the following siRNA sequences (Eurofins) were used: for hnRNP A1 sequence siA1 (5′-CAGCUGAGGAAGCUCUUCA[dT][dT]-3′) and for hnRNP A2 sequence siA2 (5′-GGAACAGUUCCGUAAGCUC[dT][dT]-3′). The siRNAs were cotransfected by mixing 177 μl Opti-MEM (Thermo Fisher Scientific) with 80 nM of each siRNA, and a second mix was composed of 17 μl Opti-MEM (Thermo Fisher Scientific) and 3 μl Oligofectamine (Thermo Fisher Scientific). The two mixes were left for 5 min at RT, and they were mixed together and left for 20 min at RT. Finally, the siRNAs were added to the cells. To obtain efficient knockdown of the proteins, a second round of silencing was performed following the same protocol. On day 2, after at least 6 h from the second round of silencing, the CFTR exon 9 C155T plasmid was transfected using Effectene (QIAGEN) reagent following the manufacturer's instructions. After 24 h, the cells were collected and prepared for the following analysis, such as the silencing rate and the CFTR exon 9 splicing. In both cases, evaluation was performed just at the RNA level, which was extracted using the RNeasy kit (QIAGEN) following the manufacturer's instructions.

Assessment of transcript levels upon knockdown was performed by real-time quantitative PCR (RT–qPCR) using PowerUp SYBR Green Master Mix (Applied Biosystems) and gene-specific primers for hnRNP A1 (5′-TCAGAGTCTCCTAAAGAGCCC-3′ sense; 5′-ACCTTGTGTGGCCTTGCAT-3′ anti-sense) and A2 (5′-TGGAGG-TAGCCCCGTTAT-3′ sense and 5′-GGACCGTAGTTAGAAGGTTGCT-3′ anti-sense). The cDNA was diluted 1:10 and subjected to 45 cycles of the following thermal protocol: 95°C for 3 min, 95°C for 10 s, 60°C for 30 s, 95°C for 10 s, 65°C for 1 s. Relative gene expression levels were determined using QuantStudio Design and Analysis Software (v1.5.1; Thermo Fisher Scientific) always comparing treated samples (siA1A2) with their direct controls (siLUC), and the normalization was performed against Gapdh. The same RNA was also used to test the CFTR exon 9 inclusion/exclusion by RT–PCR, following the protocol mentioned above. PCR products were then separated by capillary electrophoresis using QIAxcel DNA High Resolution Kit (QIAxcel), and splicing transitions were quantified using QIAxcel software (QIAxcel ScreenGel v1.4.0). The percentage of exon inclusion calculated by the software was then plotted in a column graph, and a t test analysis was performed using GraphPad Software on three independent experiments.

## Data Availability

The authors confirm that the data supporting the findings of this study are available within the article.

## Acknowledgements

We wish to thank Prof. Radovan Fiala from Josef Dadok National NMR Centre for the skillful NMR technical assistance. We acknowledge CF BIC and CF NMR, Instruct-CZ Centre, supported by MEYS CR (LM2018127) and European Regional Development Fund-Project "UP CIISB" (No. CZ.02.1.01/0.0/0.0/18_046/0015974). We thank Francesca Paron for help with splicing assays and Fig 4

and Andrea Tripepi for help with protein preparation and deprotecting the two ssRNAs for FA measurements. PJ Lukavsky acknowledges funding from a Marie Curie Action—Career Integration Grant (PCIG14-GA-2013-630758), an EMBO Installation Grant (3014), and three Czech Science Foundation Grants (P305/15/21122S, P305/18/08153S, and P305/22/20110S). E Buratti is supported by AFM Telethon (project 23788).

## Author Contributions

C Beaumont: data curation, formal analysis, validation, investigation, and visualization.

C Stuani: data curation, formal analysis, validation, investigation, and visualization.

M-Y Chou: data curation, formal analysis, validation, and investigation.

H Shakoor: data curation, formal analysis, and investigation.

M Zlobina: data curation, formal analysis, validation, and investigation.

V Palaggi: data curation, formal analysis, and investigation.

E Buratti: conceptualization and data curation.

PJ Lukavsky: conceptualization, resources, formal analysis, supervision, investigation, visualization, project administration, and writing—original draft, review, and editing.

## Conflict of Interest Statement

The authors declare that they have no conflict of interest.

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
