## [Reviewer comments · Life Science Alliance]

Life Science Alliance

hnRNP A1 induces aberrant CFTR exon 9 splicing via a newly discovered ESS element

Christelle Beaumont, Cristiana Stuani, Ming-Yuan Chou, Huma Shakoor, Maria Zlobina, Veronica Palaggi, Emanuele Buratti, and Peter Lukavsky

DOI: <https://doi.org/10.26508/lsa.202402720>

Corresponding author(s): Peter Lukavsky, Central European Institute of Technology - Masaryk University

Review Timeline:

Submission Date:	2024-03-14
Editorial Decision:	2024-05-01
Revision Received:	2025-04-23
Editorial Decision:	2025-05-28
Revision Received:	2025-06-05
Accepted:	2025-06-09

Scientific Editor: Tim Fessenden

Transaction Report:

May 1, 2024

Re: Life Science Alliance manuscript #LSA-2024-02720

Dr. Peter Josef Lukavsky
Central European Institute of Technology - Masaryk University
Kamenice 5
Brno 62500
Czech Republic

Dear Dr. Lukavsky,

Thank you for submitting your manuscript entitled "hnRNP A1 induces aberrant CFTR exon 9 splicing via a newly discovered ESS element" to Life Science Alliance. The manuscript was assessed by expert reviewers, whose comments are appended to this letter. We invite you to submit a revised manuscript addressing the Reviewer comments.

Thank you for this interesting contribution to Life Science Alliance. We are looking forward to receiving your revised manuscript.

Sincerely,

B. MANUSCRIPT ORGANIZATION AND FORMATTING:

Reviewer #1 (Comments to the Authors (Required)):

The abnormal splicing of CFTR exon 9 is influenced by RNA-protein interactions, leading to the development of a non-functional chloride channel linked to cystic fibrosis. In this study, the missplicing is linked to the recruitment of hnRNP A1 to a cis-acting element within the exon. Isothermal thermal calorimetry and NMR spectroscopy are used to characterise the relationship between hnRNP A1 the elements identified.

Overall, the data supports the conclusion that the sequence functions as exonic splicing silencers. However, it's noteworthy that disrupting just one of the hnRNPA1 binding motifs seems adequate to abolish exon 9 skipping. Consequently, I am not sure whether it can adequately be categorized as bipartite.

The authors should also exercise caution when drawing conclusions from the silencing experiment depicted in figure 3 as I believe it does not differentiate between the effects of reducing hnRNPA1 or A2 on splicing through the cis-acting elements identified in the study, and the effects resulting from the interaction of these proteins with TDP-43/A1 complex binding to the TG11-T5 repeats. One possible approach would be to experimentally inhibit the activity of these two proteins in a setting where TDP-43 is reduced. An experiment such as this may provide a more comprehensive overview of the hierarchy of these splicing silencer sequences and effects. I understand that this experiment may be beyond the scope of this paper.

Since DNA and RNA sequences may exhibit diverse structures, the use of RNA and DNA sequences in the structural tests should be explained to avoid confusing the reader

Reviewer #2 (Comments to the Authors (Required)):

In this study, Beaumont et al. investigated the interaction between hnRNP A1 and a novel exonic splicing silencer (ESS) element in CFTR pre-mRNA and studied its effects on aberrant splicing of exon 9. The authors first identified a previously unknown ESS element in exon 9 of CFTR by a minigene splicing assay with site-directed mutagenesis. Then they found that this ESS element is similar to the consensus binding sites of hnRNP A1 and hnRNP A2, and further demonstrated that knockdown of both hnRNP A1 and hnRNP A2 together increased exon 9 inclusion in another minigene assay. Finally, they determined the affinities of short ssDNA molecules (with sequences homologous to the bipartite ESS motifs) with RRM1 and RRM2 of hnRNP A1 using ITC measurements, and tried to reveal the binding patterns between the ssDNAs and RRM1/2 of hnRNP A1 using NMR spectroscopy.

Overall, this study does provide new information about the aberrant splicing of exon 9 of CFTR pre-mRNA. However, the conclusions were not fully supported by the data, due to shortcomings in the experimental design of this study.

Specific comments:

1. The newly discovered ESS element in exon 9 of CFTR pre-mRNA is an RNA element, but the authors used ssDNA molecules (instead of RNA molecules) in the ITC and NMR experiments. Although previous studies showed that hnRNP A1 binds ssDNA and RNA in a similar pattern (qualitatively speaking), there is no evidence that the binding affinities of hnRNP A1 with ssDNA and RNA are the same (quantitatively speaking). Since the ESS in exon 9 is an RNA element, it is important that the authors use RNA molecules (not ssDNA molecules) to establish the submicromolar affinities for hnRNP A1.

2. Figure 2C, it is not conclusive that the splicing changes observed in the mutant minigenes (mut4 and mut7) are specifically caused by hnRNP A1. In order to prove that, the authors need to overexpress hnRNP A1 in the minigene assay to show that exon 9 was skipped in the WT minigene but not in the mutant minigenes (mut 4 and mut7) when hnRNP A1 is overexpressed.

3. Figure 3, the authors knocked down both hnRNP A1 and hnRNP A2 together to increase exon 9 inclusion. It is not conclusive whether exon 9 inclusion is specifically caused by hnRNP A1 knockdown or by hnRNP A2 knockdown. In other words, it is not conclusive whether this ESS element is regulated by hnRNP A1 or by hnRNP A2.

4. The authors claimed that the ESS is a bipartite element, however, in the minigene assay the authors did not test whether the first motif is also important for exon 9 splicing. For example, does mutation of the first site from CAGGGAT to CAGGCCC (which still has the same 3'ss and the nucleotide at position +1) increase exon 9 inclusion? A positive result from this experiment will

significantly strengthen the conclusion of this study.

Reviewer #3 (Comments to the Authors (Required)):

The paper by Lukavsky and coworkers describes the finding of a ESS (exonic splicing enhancer) controlling the splicing of exon 9 of CFTR. The authors propose that this silencer is bound by hnRNP A1 and attempt then to characterise the mode of binding on this ESS using NMR spectroscopy. The paper has four main result sections of variable quality and impact. In part 1 (3.1), the authors do characterize the ESS by mutagenesis and splicing assays. This is quite fine but the part missing would be a direct proof that hnRNP A1 is binding the ESS which is not provided. In the next sections, the authors do a biochemical characterization of the binding of hnRNP A1 (maybe it would be better to use UP1 rather than hnRNP A1 since the whole study uses a construct with only the RRM present). In section 3.2, using ITC the author shows that UP1 can bind with 0.2 micromolar K_d the whole ESS with a 1 to 1 stoichiometry while the individual RRM binds more weakly and with a 2 to 1 protein to DNA ratio. In the next section (3.3), ITC of the individual RRM and small parts of the DNA are done to show that each RRM can bind each part of the ESS (the 5' side containing an AG sequence and the 3' side containing GGG). The 5' side not surprisingly is binding better each RRM than the 3' side since hnRNP A1 has a sequence preference for AG. The last result section (3.4) investigates using NMR the binding of these two DNA sections to the RRM using NMR. This reveals that for each DNA binding the canonical beta-sheet surface of each RRM which is not really surprising. The authors do not go beyond this point. Altogether, this is a valuable piece of research, especially the identification of the ESS but the structural work lacks some depth and some controls to make it really interesting. Here are a number of major points that would need to be addressed to make this paper suitable for publication.

Major Points:

1_ It is not clear why an NMR study of the two RRM in tandem (UP1) bound to the whole ESS is not presented. ITC data in section 3.2 shows that a 1 to 1 complex would form with fairly high affinity. With such complex, one could then identify which part of the DNA is bound to which RRM by comparing the shifts with the individual complexes of section 3.4. Also, one may also identify the path of binding and maybe able to model the complex similarly to what was done in ref 36 with ISS-N1 of the SMN2 pre-mRNA.

2_ In cells, hnRNP A1 will be binding RNA and not DNA. A control to show that the DNA binding is similar to RNA binding would be needed at least for the whole ESS bound to UP1.

3_ In section 3.4, the authors use a different DNA. One should test that the affinity is not weaker than the original DNA using ITC.

4_ The authors identify the ESS using cell assay and some mutants. Why these mutants were not used to see if the affinity for hnRNP A1 is weakened in vitro. Such ITC should be done as well.

Minor points:

1_ The authors should use U instead of T when mentioning RNA. This is important as most in vitro experiments are made with DNA, but when talking about splicing, the binding sequence is made of RNA.

2_ on page 10, section 3.2 there are many typos and on the last sentence the reference 33 is inappropriate. Maybe ref 36 would be better. In fact ref 36 should be referred to much earlier as this is really relevant to the present work as this is also a study of UP1 with an silencer of similar size.

Answers to reviewers:

Reviewer #1 (Comments to the Authors (Required)):

The abnormal splicing of CFTR exon 9 is influenced by RNA-protein interactions, leading to the development of a non-functional chloride channel linked to cystic fibrosis. In this study, the missplicing is linked to the recruitment of hnRNP A1 to a cis-acting element within the exon. Isothermal thermal calorimetry and NMR spectroscopy are used to characterize the relationship between hnRNP A1 the elements identified.

Overall, the data supports the conclusion that the sequence functions as exonic splicing silencers. However, it's noteworthy that disrupting just one of the hnRNPA1 binding motifs seems adequate to abolish exon 9 skipping. Consequently, I am not sure whether it can adequately be categorized as bipartite.

- All binding and NMR experiments indicate that the bipartite ESS interacts with both RRM1 and RRM2 of hnRNP A1 and that the binding is cooperative.

The authors should also exercise caution when drawing conclusions from the silencing experiment depicted in figure 3 as I believe it does not differentiate between the effects of reducing hnRNPA1 or A2 on splicing through the cis-acting elements identified in the study, and the effects resulting from the interaction of these proteins with TDP-43/A1 complex binding to the TG11-T5 repeats. One possible approach would be to experimentally inhibit the activity of these two proteins in a setting where TDP-43 is reduced. An experiment such as this may provide a more comprehensive overview of the hierarchy of these splicing silencer sequences and effects. I understand that this experiment may be beyond the scope of this paper.

- The Buratti group has shown over several years that TDP-43 knockdown completely abolishes exon 9 skipping; thus it would be difficult to study the effect of hnRNP A1 and the ESS on exon 9 skipping under TDP-43 knockdown conditions.

Since DNA and RNA sequences may exhibit diverse structures, the use of RNA and DNA sequences in the structural tests should be explained to avoid confusing the reader

- Throughout the text, we now clearly indicate whether ssRNA or ssDNA was used.

Reviewer #2 (Comments to the Authors (Required)):

In this study, Beaumont et al. investigated the interaction between hnRNP A1 and a novel exonic splicing silencer (ESS) element in CFTR pre-mRNA and studied its effects on aberrant splicing of exon 9. The authors first identified a previously unknown ESS element in exon 9 of CFTR by a minigene splicing assay with site-directed mutagenesis. Then they found that this ESS element is similar to the consensus binding sites of hnRNP A1 and hnRNP A2, and further demonstrated that knockdown of both hnRNP A1 and hnRNP A2 together increased exon 9 inclusion in another minigene assay. Finally, they determined the affinities of short ssDNA molecules (with sequences homologous to the bipartite ESS motifs) with RRM1 and RRM2 of hnRNP A1 using ITC measurements, and tried to reveal the binding patterns between the ssDNAs and RRM1/2 of hnRNP A1 using NMR spectroscopy.

Overall, this study does provide new information about the aberrant splicing of exon 9 of CFTR pre-mRNA. However, the conclusions were not fully supported by the data, due to shortcomings in the experimental design of this study.

Specific comments:

1. The newly discovered ESS element in exon 9 of CFTR pre-mRNA is an RNA element, but the authors used ssDNA molecules (instead of RNA molecules) in the ITC and NMR experiments. Although previous studies showed that hnRNP A1 binds ssDNA and RNA in a similar pattern (qualitatively speaking), there is no evidence that the binding affinities of hnRNP A1 with ssDNA and RNA are the same (quantitatively speaking). Since the ESS in exon 9 is an RNA element, it is important that the authors use RNA molecules (not ssDNA molecules) to establish the submicromolar affinities for hnRNP A1.

- We explain now in the text why ITC experiments with ssRNA were not successful. We performed FA measurements instead with wt and mut4 ESS ssRNA which reveals binding in the nanomolar range similar to ssDNA and a weakened binding for the mut4 ssRNA (new Fig 5).

2. Figure 2C, it is not conclusive that the splicing changes observed in the mutant minigenes (mut4 and mut7) are specifically caused by hnRNP A1. In order to prove that, the authors need to overexpress hnRNP A1 in the minigene assay to show that exon 9 was skipped in the WT minigene but not in the mutant minigenes (mut 4 and mut7) when hnRNP A1 is overexpressed.

- We performed the minigene assay with overexpression of hnRNP A1 and show that the effects of the overexpression is abolished in the mutants that carry mutations in the ESS (new Fig 3).

3. Figure 3, the authors knocked down both hnRNP A1 and hnRNP A2 together to increase exon 9 inclusion. It is not conclusive whether exon 9 inclusion is specifically caused by hnRNP A1 knockdown or by hnRNP A2 knockdown. In other words, it is not conclusive whether this ESS element is regulated by hnRNP A1 or by hnRNP A2.

- We explain the reason for double kd in the text now:
"Finally, in order to further determine whether hnRNP A1 could act as splicing silencer proteins of CFTR exon 9, we silenced both hnRNP A1 and A2 proteins in the presence of a CFTR exon 9 minigene which also carried the C155T artificial variant. This variant was designed in previous studies to display a higher level of exon 9 skipping (approximately 50%) in order to better appreciate eventual changes in exon 9 inclusion in both directions (Pagani, Buratti, Stuardi, & Baralle, 2003) and is therefore ideal to test for variations in splicing factor expression levels. The reason why hnRNP A1 and A2 had to be silenced together is due to the fact that both proteins are well known to compensate for each other in many pre-mRNA splicing events including CFTR exon 9 (D'Ambrogio et al., 2009)."

4. The authors claimed that the ESS is a bipartite element, however, in the minigene assay the authors did not test whether the first motif is also important for exon 9 splicing. For example, does mutation of the first site from CAGGGAT to CAGGCC (which still has the same 3'ss and the nucleotide at position

+1) increase exon 9 inclusion? A positive result from this experiment will significantly strengthen the conclusion of this study.

- We show that the binding of hnRNP A1 tandem RRM is cooperative in the nanomolar range while the individual RRM binds the separated bipartite ESS motif with micromolar affinity. These results are also consistent with the NMR titration experiments where the individual RRM binds both motifs.

Reviewer #3 (Comments to the Authors (Required)):

The paper by Lukavsky and coworkers describes the finding of a ESS (exonic splicing enhancer) controlling the splicing of exon 9 of CFTR. The authors propose that this silencer is bound by hnRNP A1 and attempt then to characterize the mode of binding on this ESS using NMR spectroscopy. The paper has four main result sections of variable quality and impact. In part 1 (3.1), the authors do characterize the ESS by mutagenesis and splicing assays. This is quite fine but the part missing would be a direct proof that hnRNP A1 is binding the ESS which is not provided. In the next sections, the authors do a biochemical characterization of the binding of hnRNP A1 (maybe it would be better to use UP1 rather than hnRNP A1 since the whole study uses a construct with only the RRM present). In section 3.2, using ITC the author shows that UP1 can bind with 0.2 micromolar K_d the whole ESS with a 1 to 1 stoichiometry while the individual RRM binds more weakly and with a 2 to 1 protein to DNA ratio. In the next section (3.3), ITC of the individual RRM and small parts of the DNA are done to show that each RRM can bind each part of the ESS (the 5' side containing an AG sequence and the 3' side containing GGG). The 5' side not surprisingly is binding better each RRM than the 3' side since hnRNP A1 has a sequence preference for AG. The last result section (3.4) investigate using NMR the binding of these two DNA sections to the RRM using NMR. This reveals that for each DNA binding the canonical β -sheet surface of each RRM which is not really surprising. The authors do not go beyond this point.

Altogether, this is a valuable piece of research, especially the identification of the ESS but the structural work lacks some depth and some controls to make it really interesting. Here are a number of major points that would need to be addressed to make this paper suitable for publication.

Major Points:

1_ It is not clear why an NMR study of the two RRM in tandem (UP1) bound to the whole ESS is not presented. ITC data in section 3.2 shows that a 1 to 1 complex would form with fairly high affinity. With such complex, one could then identify which part of the DNA is bound to which RRM by comparing the shifts with the individual complexes of section 3.4. Also, one may also identify the path of binding and maybe able to model the complex similarly to what was done in ref 36 with ISS-N1 of the SMN2 pre-mRNA.

- NMR data were of really poor quality, most of the peaks at the binding site were lost both with ssDNA (3 constructs) and ssRNA (4 constructs) making a firm interpretation impossible.

2_ In cells, hnRNP A1 will be binding RNA and not DNA. A control to show that the DNA binding is similar to RNA binding would be needed at least for the whole ESS bound to UP1.

- We explain now in the text why ITC experiments with ssRNA were not successful. We performed FA measurements instead with wt and mut4 ESS ssRNA which reveals binding in the nanomolar range similar to ssDNA and a weakened binding for the mut4 ssRNA (new Fig 5).

3_ In section 3.4, the authors use a different DNA. One should test that the affinity is not weaker than the original DNA using ITC.

- The change in sequence has no effect on splicing (mut1 in fig2C) and does not alter the AG-motif. We therefore did not test affinity since the nucleotide change has no biological affect.

4_ The authors identify the ESS using cell assay and some mutants. Why these mutants were not used to see if the affinity for hnRNP A1 is weakened in vitro. Such ITC should be done as well.

- We explain now in the text why ITC experiments with ssRNA were not successful. We performed FA measurements instead with wt and mut4 ESS ssRNA which reveals binding in the nanomolar range similar to ssDNA and a weakened binding for the mut4 ssRNA (new Fig 5).

Minor points:

1_ The authors should use U instead of T when mentioning RNA. This is important as most in vitro experiment are made with DNA, but when talking about splicing, the binding sequence is made of RNA.

- We now use U instead of T when mentioning RNA instead of DNA.

2_ on page 10, section 3.2 there many typos and on the last sentence the reference 33 is inappropriate. Maybe ref 36 would be better. In fact ref 36 should be referred to much earlier as this is really relevant to the present work as this is also a study of UP1 with an silencer of similar size.

- The Allain data and reference is now moved to the introduction and mentioned again in the discussion. It is not clear to us why ref 33 is inappropriate, please, explain.

May 28, 2025

RE: Life Science Alliance Manuscript #LSA-2024-02720R

Dr. Peter Josef Lukavsky
Central European Institute of Technology - Masaryk University
Kamenice 5
Brno 62500
Czech Republic

Dear Dr. Lukavsky,

Thank you for submitting your revised manuscript entitled "hnRNP A1 induces aberrant CFTR exon 9 splicing via a newly discovered ESS element". As you will see below, all reviewers are satisfied with the revised manuscript. We would be happy to publish your paper in Life Science Alliance pending final revisions necessary to meet our formatting guidelines.

- Please upload your main and supplementary figures as single files.
- Please add the X and Bluesky handles of your host institute/organization, as well as your own and one of the authors, to our system.
- Please be sure that the authorship listing and order are correct and match between the manuscript file and the system.
- It is recommended to exclude figures from the manuscript text and upload them separately.
- Please add your figure legends to the main manuscript text after the references section.
- Please use the [10 author names, et al.] format in your references (i.e., limit the author names to the first 10).
- Please add callouts for Figures 3A-B; 8A and C to your main manuscript text.

A. FINAL FILES:

B. MANUSCRIPT ORGANIZATION AND FORMATTING:

per figure for this information. These files will be linked as supplementary "Source Data" files.

Sincerely,

Reviewer #1 (Comments to the Authors (Required)):

The authors have done a considerable effort to reply to my queries. I believe the manuscript can now be published

Reviewer #2 (Comments to the Authors (Required)):

In this revision, the authors have adequately addressed all of the reviewers' comments. The manuscript is suitable for publication in its current form.

Reviewer #3 (Comments to the Authors (Required)):

I am now happy with the proposed changes and the answers to the criticisms.

June 9, 2025

RE: Life Science Alliance Manuscript #LSA-2024-02720RR

Dr. Peter Josef Lukavsky
Central European Institute of Technology - Masaryk University
Gorkého 76/31
Brno 60200
Czech Republic

Dear Dr. Lukavsky,

Thank you for submitting your Research Article entitled "hnRNP A1 induces aberrant CFTR exon 9 splicing via a newly discovered ESS element". It is a pleasure to let you know that your manuscript is now accepted for publication in Life Science Alliance. Congratulations on this interesting work.

DISTRIBUTION OF MATERIALS:

Again, congratulations on a very nice paper. I hope you found the review process to be constructive and are pleased with how the manuscript was handled editorially. We look forward to future exciting submissions from your lab.

Sincerely,
